# Uncertainty-inspired open set learning for retinal anomaly identification

Meng Wang [1,14], Tian Lin [2,14], Lianyu Wang[3,4,14], Aidi Lin[2], Ke Zou[5], Xinxing Xu [1], Yi Zhou[6], Yuanyuan Peng[7], Qingquan Meng[6], Yiming Qian [1], Guoyao Deng[5], Zhiqun Wu[8], Junhong Chen[9], Jianhong Lin[10], Mingzhi Zhang [2], Weifang Zhu[6], Changqing Zhang[11], Daoqiang Zhang [3,4], Rick Siow Mong Goh[1], Yong Liu[1], Chi Pui Pang[2,12], Xinjian Chen [6,13,15] ✉, Haoyu Chen [2,15] ✉ & Huazhu Fu [1,15] ✉

Failure to recognize samples from the classes unseen during training is a major limitation of artificial intelligence in the real-world implementation for recognition and classification of retinal anomalies. We establish an uncertainty-inspired open set (UIOS) model, which is trained with fundus images of 9 retinal conditions. Besides assessing the probability of each category, UIOS also calculates an uncertainty score to express its confidence. Our UIOS model with thresholding strategy achieves an F1 score of 99.55%, 97.01% and 91.91% for the internal testing set, external target categories (TC)-JSIEC dataset and TC-unseen testing set, respectively, compared to the F1 score of 92.20%, 80.69% and 64.74% by the standard AI model. Furthermore, UIOS correctly predicts high uncertainty scores, which would prompt the need for a manual check in the datasets of non-target categories retinal diseases, low-quality fundus images, and non-fundus images. UIOS provides a robust method for real-world screening of retinal anomalies.

Retina is part of the central nervous system responsible for phototransduction. Retinal diseases are the leading cause of irreversible blindness and visual impairment worldwide. Treatment at the early stage of disease is important to reduce serious and permanent damages. Therefore, timely diagnosis and appropriate treatment are important for preventing threatened vision and even irreversible blindness. Diagnosis of retinal diseases requires expertise of trained ophthalmologists, while there is always heavy demand for large number of patients with retinal diseases to limited number of specialists. A solution to this service gap is image-based screening that alleviates workload of ophthalmologists. Fundus photography-based screening has been shown to be successful to

[1]Institute of High Performance Computing (IHPC), Agency for Science, Technology and Research (A*STAR), 1 Fusionopolis Way, #16-16 Connexis, Singapore 138632, Republic of Singapore. [2]Joint Shantou International Eye Center, Shantou University and the Chinese University of Hong Kong, 515041 Shantou, Guangdong, China. [3]College of Computer Science and Technology, Nanjing University of Aeronautics and Astronautics, 211100 Nanjing, Jiangsu, China. [4]Laboratory of Brain-Machine Intelligence Technology, Ministry of Education Nanjing University of Aeronautics and Astronautics, 211106 Nanjing, Jiangsu, China. [5]National Key Laboratory of Fundamental Science on Synthetic Vision and the College of Computer Science, Sichuan University, 610065 Chengdu, Sichuan, China. [6]School of Electronics and Information Engineering, Soochow University, 215006 Suzhou, Jiangsu, China. [7]School of Biomedical Engineering, Anhui Medical University, 230032 Hefei, Anhui, China. [8]Longchuan People's Hospital, 517300 Heyuan, Guangdong, China. [9]Puning People's Hospital, 515300 Jieyang, Guangdong, China. [10]Haifeng PengPai Memory Hospital, 516400 Shanwei, Guangdong, China. [11]College of Intelligence and Computing, Tianjin University, 300350 Tianjin, China. [12]Department of Ophthalmology and Visual Sciences, The Chinese University of Hong Kong, 999077 Hong Kong, China. [13]State Key Laboratory of Radiation Medicine and Protection, Soochow University, 215006 Suzhou, China. [14]These authors contributed equally: Meng Wang, Tian Lin, Lianyu Wang. [15]These authors jointly supervised this work: Xinjian Chen, Haoyu Chen, Huazhu Fu. ✉e-mail: xjchen@suda.edu.cn; drchenhaoyu@gmail.com; hzfu@ieee.org

prevent irreversible vision impairment and blindness caused by diabetic retinopathy[1].

In recent years, deep learning, as an established but still rapidly evolving technology, has remarkably enhanced disease screening from medical imaging[2–4], including fundus photography screening for retinal diseases. The applications of deep learning in diabetic retinopathy (DR)[5–8], glaucoma[9–11], and age-related macular degeneration (AMD)[12–14] screening have achieved comparable performance with human experts. There are also some successful applications of deep learning in classifying multiple retinal diseases[15].

However, a major drawback of the standard artificial intelligence (AI) models in real-world implementation is the problem of open set recognition. AI models are trained in a close set, i.e., a limited number of categories and limited characters of samples. But the real world is an open set environment, where some samples may be out of the categories in the training set or with untypical features. Previous studies have demonstrated that the performance of deep learning models declines when applied to data out of distribution (OOD), such as low-quality images and untypical cases[16–18]. Furthermore, if the testing image is a retinal disease not included in the training set, even if it is a non-fundus image, the standard AI model will still give a diagnosis of the disease category in the training data. This would lead to misdiagnosis. Meanwhile, in practice, it is impossible to collect data that covers all fundus abnormalities with sufficient sample size to train the model. Therefore, it is highly necessary to develop an open set learning model that can accurately classify retinal diseases included in the training set, as well as for the screening of other OOD samples without the need to collect and label additional data.

In this study, we developed a fundamental AI model of uncertainty-inspired open set (UIOS) based on the evidential uncertainty deep neural network. As shown in Fig. 1, if the test data is a fundus disease included in the training set with distinct features, our proposed UIOS model will give a diagnosis decision with a low uncertainty score, which indicates that the decision is reliable. On the contrary, if the test data is in the category outside the training data set, low-quality images, and non-fundus data, UIOS will give a prediction result with a high uncertainty score, which suggests that the diagnosis result given by the AI model is unreliable. If so, a manual check by an

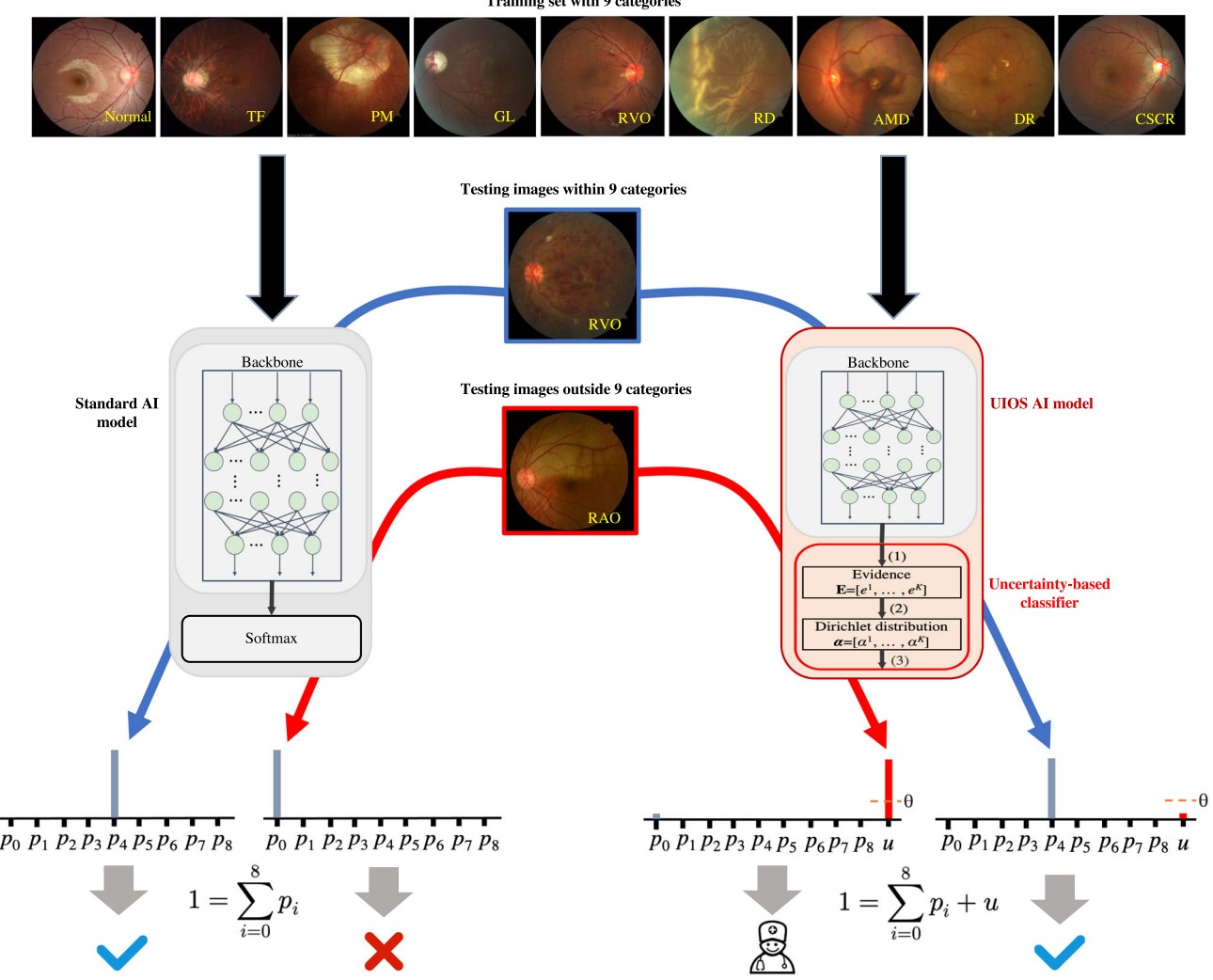

**Fig. 1 | The overview of the uncertainty-inspired open set (UIOS) learning for retinal anomaly classification.** Standard artificial intelligence (AI) and our proposed UIOS AI models were trained with the same dataset with 9 categories of retinal photos. In testing, standard AI model assigns a probability value ($p_i$) to each of the 9 categories, and the one with the highest probability is output as the diagnosis. Even when the model is tested with a retinal image with disease outside the training set, the model still outputs one from the 9 categories, which may lead to misdiagnosis. In contrast, UIOS outputs an uncertainty score ($\mu$) besides the probability ($p_i$) for the 9 categories. When the model is fed with an image with obvious features of retinal disease in the 9 categories, the uncertainty-based classifier will output a prediction result with a low uncertainty score below the threshold $\theta$ to indicate that the diagnosis result is reliable. Conversely, when the input data contains ambiguous features or is an anomaly outside of training categories, the model will assign a high uncertainty score above threshold $\theta$ to explicitly indicate that the prediction result is unreliable and requires a double-check from their ophthalmologist to avoid misdiagnosis.

experienced grader or ophthalmologist is required. Therefore, with the estimated uncertainty, our AI model is capable to give reliable diagnosis for retinal diseases involved in training data and avoid confusion from OOD samples.

## Results

### Performance in the internal testing dataset

In the internal testing set with 2010 images, our UIOS achieved an F1 score ranging from 93.12% to 99.27% for the 9 categories, especially for pathologic myopia (PM, 98.84%), glaucoma (GL, 98.53%), retinal detachment (RD, 99.27%), and diabetic retinopathy (DR, 98.04%) (Table 1). The average area under the curve (AUC) (Fig. 2), precision (Supplementary Table 1), F1 score (Table 1), sensitivity (Supplementary Table 2), and specificity (Supplementary Table 3) of the UIOS model were 99.79%, 97.57%, 97.29%, 97.04%, and 99.75%, respectively, which were better than the standard AI model, although the difference was statistically significant for F1 ($p = 0.029$, Supplementary Table 7) but not AUC ($p = 0.371$, Supplementary Table 8). Furthermore, UIOS also outperformed the standard AI model in terms of confusion matrix (Supplementary Fig. 1). It should be noted that when an image is flagged as "uncertain" beyond the threshold by the UIOS model, those images are suggested to seek double checking by ophthalmologists and removed when calculating the eventual diagnostic performance metrics.

The distribution of the uncertainty score in the primary testing set was similar to the validation set, except that 9.75% of samples with uncertainty scores were above the threshold (Fig. 3 and Supplementary Table 4). After thresholding these OOD samples, the performance of UIOS was further improved. The average value of all indicators has reached more than 99%, especially the average F1 score and AUC were 99.55% and 99.89%, respectively with the UIOS+thresholding (Table 1 and Fig. 2c).

In addition, we compared the performance of UIOS with other commonly used uncertainty methods, including Monte Carlo drop-out (MC-Drop), ensemble models (Ensemble), test time-augmentations (TTA), and using entropy across the categorical class probabilities in the standard AI model (Entropy). Our UIOS model consistently outperformed these uncertainty approaches in terms of F1 score, both on the original internal testing set (Supplementary Table 5) and dataset where samples with uncertainty scores above their threshold have been suggested to seek double-checking by ophthalmologists (Supplementary Table 6). Statistical analyses showed that the difference was significant except in the comparison of UIOS to Ensemble in the internal testing set with thresholding (Supplementary Table 7). The

receiver operating characteristic (ROC) curves of different uncertainty methods are shown in Supplementary Figs. 2 and 3, and the statistical analyses are shown in Supplementary Table 8. The AUCs of UIOS are higher or comparable in performance to other methods.

### Performance in the external datasets

To further evaluate the generality of UIOS for screening fundus diseases, we also conducted experiments on two external datasets of target categories from JSIEC1000 (TC-JSIEC) and unseen target categories (TC-unseen), with 435 and 3,716 images, respectively. Both external datasets had the same categories as the training set. The TC-JSIEC set was from a different source, while the images in the TC-unseen dataset have different features, such as early stage or ambiguous features. The performance of the standard AI model declined in these models and achieved an average F1 score of 80.69% and 64.74%, respectively (Table 1). In comparison, UIOS achieved an average F1 score of 87.19% and 77.15%, with a $p$ value of 0.006 and 0.008, respectively, for the comparison with standard AI model (Table 1 and Supplementary Table 7). The improvement of the F1 score was found in all categories (Table 1).

There were 23.22% and 47.55% samples with an uncertainty score over the threshold, in the TC-JSIEC and TC-unseen sets, respectively (Fig. 4 and Supplementary Tables 4 and 9), which indicated the need for assessment by ophthalmologists. After thresholding these samples, the F1 of UIOS was further improved from 87.19% to 97.01% and from 77.15% to 91.91%, respectively (Table 1). The precision, sensitivity, and specificity were also best in the UIOS with thresholding strategy among the three models (Supplementary Tables 1–3).

The ROC curves of the three models in detecting retinal diseases in TC-JSIEC and TC-unseen datasets are shown in Fig. 2d–i. The AUC of the standard AI model was 97.67% and 91.84% for the TC-JSIEC and TC-unseen datasets, respectively. They improved to 99.07% and 93.87% with the UIOS model ($p = 0.002$ and 0.196, respectively) and further achieved 99.77% and 97.34% with the UIOS+thresholding. And, our UIOS also achieved better confusion matrices than the standard AI models on two external test sets (Supplementary Fig. 1). Furthermore, when applying our thresholding strategy (UIOS+thresholding) to indicate samples with uncertainty scores above the threshold that required manual check by ophthalmologists, we observed a further significant improvement in the confusion matrix and a significant reduction in misclassified samples (Supplementary Fig. 1).

Figure 4 shows four samples of fundus images detected with the standard AI model and our UIOS model. The standard AI model directly took the fundus category that obtained the maximum probability

**Table 1 | F1 score of different models on three testing sets**

| Category | Internal testing dataset | | | TC-JSIEC | | | TC-unseen | | |
|---|---|---|---|---|---|---|---|---|---|
| | Standard AI model | UIOS model | UIOS+ thresholding | Standard AI model | UIOS model | UIOS+ thresholding | Standard AI model | UIOS model | UIOS+ thresholding |
| Normal | 97.48 | 99.18 | 99.88 | 72.50 | 84.34 | 90.00 | 75.39 | 83.17 | 92.86 |
| TF | 93.05 | 93.12 | 98.68 | 75.86 | 78.79 | 94.74 | 59.36 | 78.43 | 89.14 |
| PM | 95.98 | 98.84 | 99.39 | 99.08 | 100.00 | 100.00 | 79.90 | 80.00 | 94.69 |
| GL | 97.26 | 98.53 | 100.00 | 60.87 | 72.73 | 93.33 | 77.69 | 78.33 | 95.08 |
| RVO | 95.72 | 97.36 | 99.60 | 86.21 | 95.24 | 100.00 | 65.48 | 84.96 | 97.03 |
| RD | 93.43 | 99.27 | 100.00 | 97.35 | 94.44 | 98.85 | 48.95 | 72.19 | 92.59 |
| AMD | 87.97 | 97.24 | 99.41 | 83.53 | 93.67 | 99.31 | 42.78 | 50.17 | 76.63 |
| DR | 93.25 | 98.04 | 99.62 | 82.54 | 87.76 | 96.83 | 53.43 | 83.21 | 96.04 |
| CSCR | 75.65 | 94.05 | 99.33 | 68.29 | 77.78 | 100.00 | 79.65 | 83.84 | 93.12 |
| Average | 92.20 | 97.29 | 99.55 | 80.69 | 87.19 | 97.01 | 64.74 | 77.15 | 91.91 |

*TF* tigroid fundus, *PM* pathological myopia, *GL* glaucoma, *RVO* retinal vein occlusion, *RD* retinal detachment, *AMD* age-related macular degeneration, *DR* diabetic retinopathy, *CSCR* central serous chorioretinopathy.

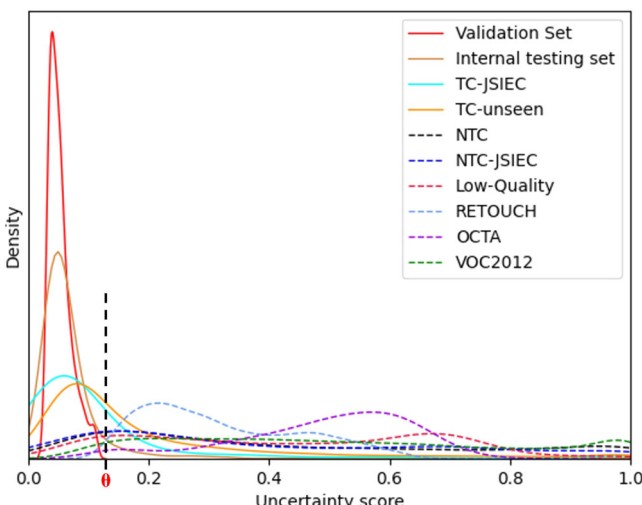

**Fig. 2 | The receiver operating characteristic (ROC) curves of the standard AI model, our UIOS, and UIOS+thresholding in internal and two external testing datasets.** Source data are provided as a Source data file.

**Fig. 3 | Uncertainty density distribution for different datasets.** Different colored solid lines indicate different test datasets for target categories of retinal diseases, while different colored dashed lines indicate different out of distribution datasets. $\theta$ threshold theta. Source data are provided as a Source data file.

value as the final diagnosis. UIOS could give the final prediction result while providing an uncertainty score to explicitly illustrate the reliability of the diagnosis. The images with lower uncertainty scores indicated higher confidence in the final decision of the model (Fig. 4a, b). In some images with incorrect final diagnosis (Fig. 4c, d), the standard AI model not only gave wrong prediction results, but also provided a higher probability value which led to mis-/under-diagnosis. In contrast, although UIOS could also gave wrong diagnostic results, the prediction results were indicated to be unreliable by assigning a high uncertainty score to the diagnostic results. The high uncertainty score suggested the need to seek an ophthalmologist to read the images again to prevent mis-/under-diagnosis.

We further compared the performance of our proposed UIOS to other uncertainty approaches in these two external testing sets. The results showed that our UIOS model achieved higher F1 scores (Supplementary Tables 10–13) and AUC (Supplementary Figs. 2 and 3) in both original datasets and the datasets after thresholding. The difference was statistically significant in most comparisons (Supplementary Tables 7 and 8).

## Open set anomaly detection

In three fundus photo datasets with abnormal samples outside the training category, UIOS detected 86.67%, 82.27% and 89.40% of samples

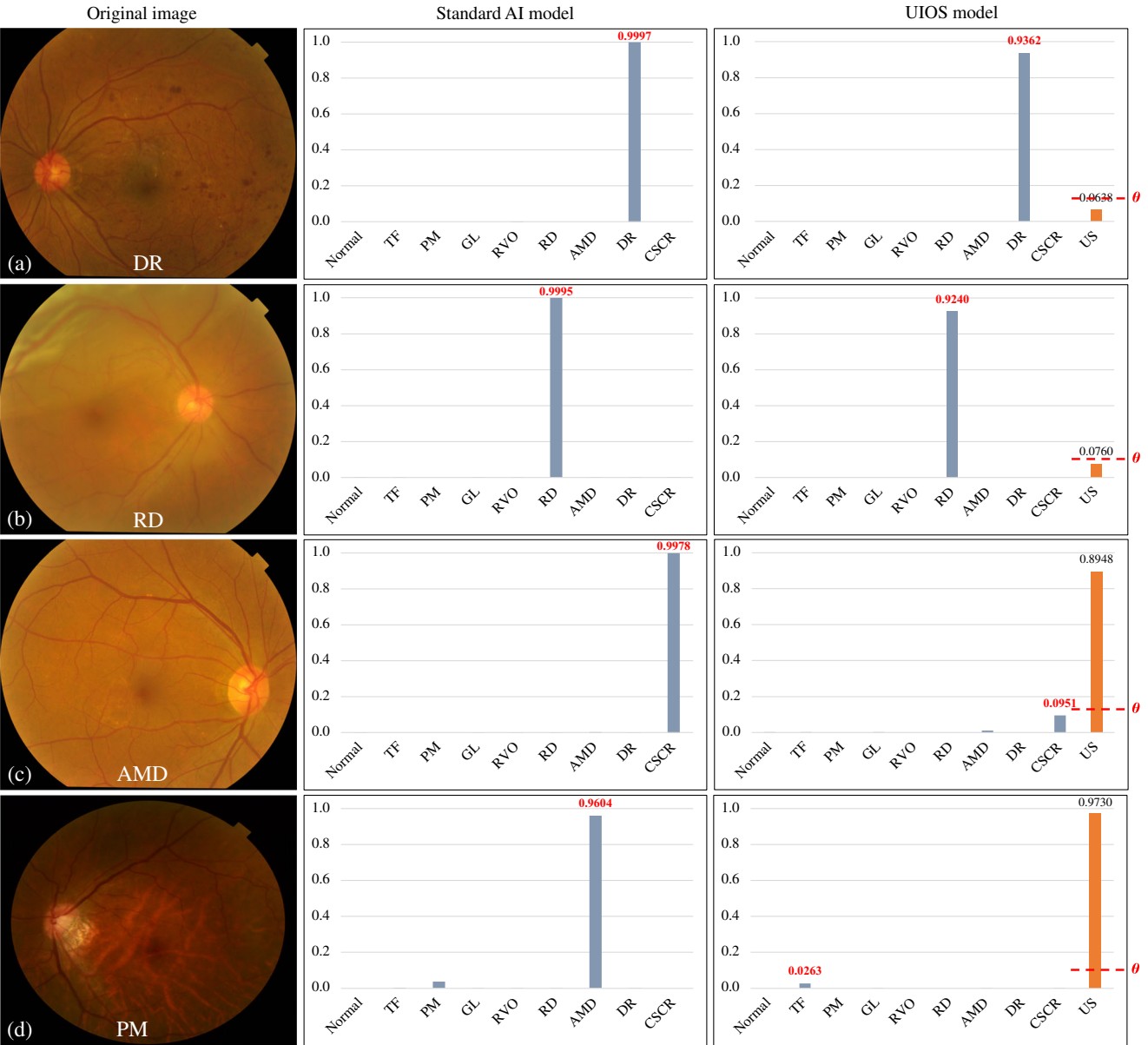

**Fig. 4 | Four samples of fundus images detected with the standard AI model and our UIOS model. a**, **b** Two samples with correct diagnostic results from both the standard AI model and our UIOS model. **c**, **d** Two samples with incorrect diagnostic results from the standard AI model and our UIOS model. Unlike the standard AI model, which directly takes the fundus disease category with the highest probability score as the final diagnosis result, our UIOS will not only give the probability scores but also provide the corresponding uncertainty score to reflect the reliability of the prediction result. If the uncertainty score is less than the threshold theta, indicating the model prediction is reliable; Conversely, if the uncertainty score is greater than the threshold theta, which represents that the result is unreliable, and manual double-checking is required to avoid possible misdiagnosis problems. US uncertainty score, $\theta$ threshold theta. Source data are provided as a Source data file.

with high uncertainty on non-target categories (NTC) dataset (1380 samples), NTC-JSIEC (502 samples) and low-quality image dataset (1066 samples), respectively. UIOS also performed well in detecting OOD samples from three non-fundus data. Specifically, UIOS achieved abnormality detection rates of 99.81%, 99.01% and 96.18% on the three non-fundus datasets RETOUCH [6396 optical coherence tomography (OCT) images of training set], OCTA [304 optical coherence tomography angiography (OCTA) images] and VOC 2012 (17,125 natural images of training and validation sets including 21 categories), respectively. Meanwhile, Fig. 3 shows the uncertainty density distribution of different datasets outside the training set category. Compared to the uncertainty score distribution of the validation set, UIOS assigned a higher uncertainty score for the samples in different OOD datasets. In addition, Fig. 5 represents some examples of OOD images that were not included in the training category. The standard AI model provided incorrect diagnosis results and assigned a high probability to the wrongly diagnosed fundus disease. Conversely, although our UIOS model gave incorrect predictions for OOD samples, it also assigned a higher uncertainty score to indicate that the final decision was unreliable and needed assessment by an ophthalmologist.

The abnormal detection rates of different uncertainty methods on different datasets are shown in Supplementary Table 14. Overall, UIOS achieved the highest anomaly detection rates on most datasets, except in the NTC-JSIEC and OCTA datasets, where UIOS was slightly lower than Entropy and Ensemble respectively. Furthermore, our UIOS model only required a single forward pass of the model to obtain uncertainty estimates, resulting in the highest execution efficiency. Particularly when compared to MC-Drop, Ensemble, and TTA methods,

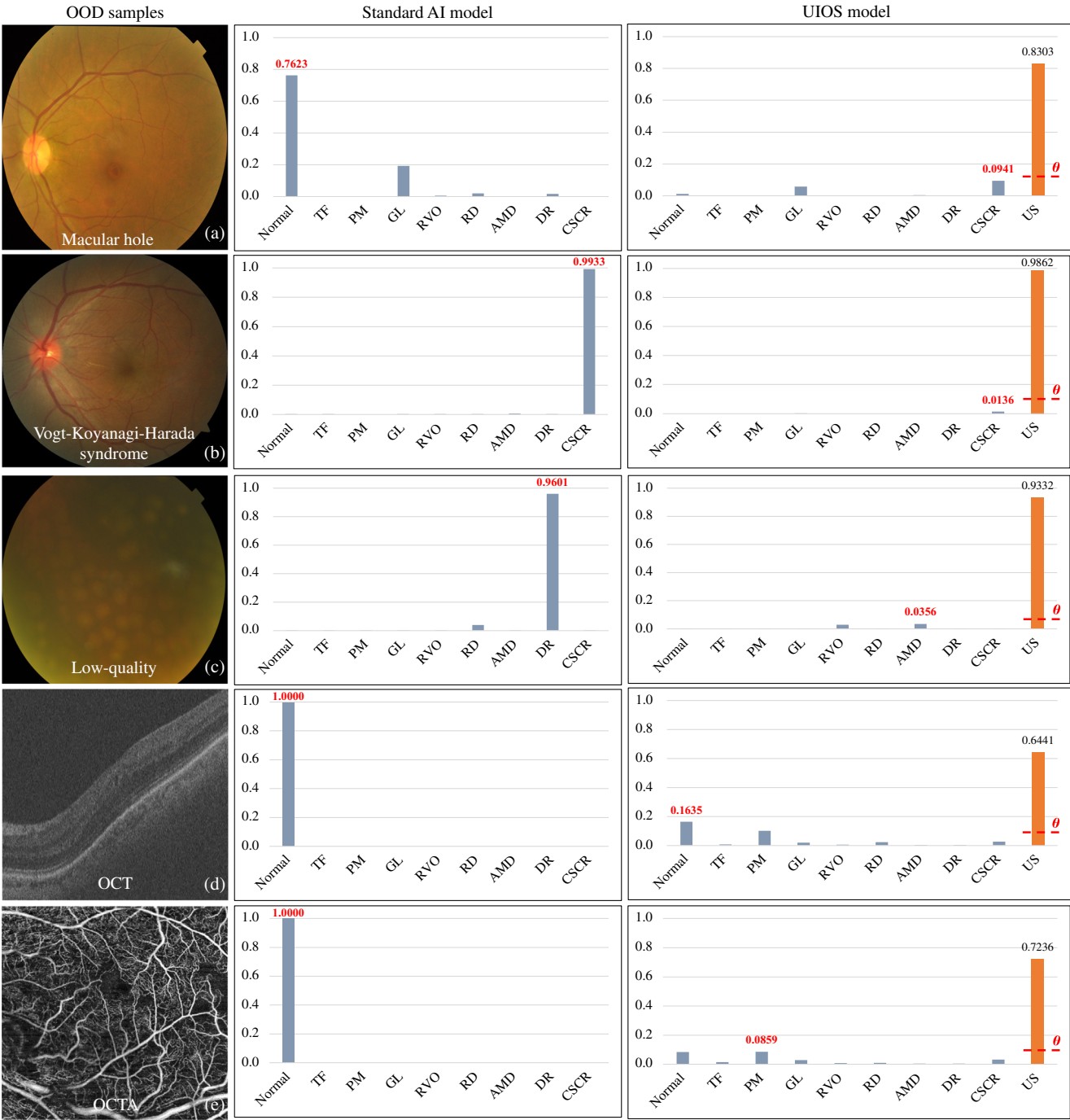

**Fig. 5 | Testing results of OOD samples that were not included in the training category.** Besides assigning a probability to OOD samples as the standard AI model, our UIOS model also assigns a high uncertainty score to indicate that the final decision is unreliable and needs a double-check. US uncertainty score, $\theta$ threshold theta. Source data are provided as a Source data file. **a** from NTC dataset, **b** also from NTC dataset, **c** from low-quality dataset, **d** an OCT image from RETOUCH dataset, and **e** an OCTA image from OCTA dataset.

UIOS showed a significant improvement in execution efficiency, with only 0.34 ms/per image (Supplementary Table 14).

## Discussion

In the past few years, deep learning-based methods for the detection of retinal diseases have shown a rapid growing trend[13–15]. But less works have been reported to address the confidence and reliability of results. Besides, AI models would inevitably give wrong prediction results for rare retinal diseases or other OOD data that are not included in the training set. While we can also retrain the model to detect more abnormal classes by collecting and labeling more categories of data, it incurs more time and labor that are costly. In addition, due to limitations of medical resources and large number of patients with different fundus diseases, it is almost impossible to collect and label all the data on retinal abnormalities. This is a major reason that limits the deployment of AI models in clinical practice. To address these issues, we provide a uncertainty-based open set AI model for retinal disease detection. We introduce an algorithm that divides the diagnostic results of the AI model into low and high confidence levels by uncertainty thresholding, which can significantly improve the accuracy of screening for target-categories fundus diseases in training set with obvious features, while also avoiding misdiagnosis due to ambiguous features. Our uncertainty

thresholding approach can detect abnormal samples to avoid incorrect diagnosis and subsequent incidences when deploying AI models in clinical practice due to samples outside the training distribution. In addition, our proposed uncertainty paradigm is highly scalable and can be combined with and enhance the performance of current commonly used baseline models for retinal diseases screening.

Recently, numerous methods have been developed to detect abnormalities in fundus images using various deep neural networks[19–22]. They trained the models with normal images only and detected abnormal images in the testing set. Although they have achieved an AUC of 0.8–0.9, these methods can only differentiate abnormal from normal images, but cannot classify abnormal images into different categories. Our UIOS model was developed based on multiple categories classification, including normal conditions, 8 retinal diseases, and other abnormalities. Therefore, UIOS should be adequate and ready for clinical implementation.

Several techniques have been explored to evaluate uncertainty from AI models. Bayesian neural network (BNNs)[18,23–25] is a common uncertainty quantification approach, which can evaluate the uncertainties in their prediction. Within BNNS, MC-Drop[26] is a more scalable and commonly used method that is achieved by randomly removing a portion of nodes from the model structure when generating predictions, which also leads to higher computational costs. Deep ensemble is another uncertainty method[27,28] which generates multiple prediction distributions by training several independent deep learning models on the same input samples and calculates the mean and variance of these distributions, where mean and variance are used as the final prediction and uncertainty. Besides, some studies explored the uncertainty evaluation based on the test time augmentation approach[29], where an input sample undergoes a series of different augmentation operations, and then the uncertainty is estimated based on the variance of prediction results from the augmented images. While there have been works exploring the application of uncertainty to medical imaging with promising performance, most of these works are based on Bayesian uncertainty and few of them are for multi-target detection of fundus images. Furthermore, there are previous works to evaluate the reliability of classification results by using entropy across the categorical class probabilities[30,31]. While entropy is effective in capturing uncertainty within the observed classes, it may not perform well when faced with out-of-distribution examples. OOD samples can have low entropy values, leading to high confidence predictions that are incorrect. Consequently, relying solely on entropy may not provide robust detection or handling of out-of-distribution data. Evidential-based subjective logistic uncertainty to calculate the uncertainty score is directly based on the evidence collected from the feature extractor network[32–34]. The potential capacity of subjective logistic to estimate the reliability of classification has been explored by Han et al.[33], who introduced Dirichlet distribution into subjective logical (SL) to derive probabilities of different classes and the overall uncertainty score. However, they have not explored how to detect OOD samples based on uncertainty in a quantitative approach. Our previous studies have introduced evidential uncertainty to investigate uncertainty estimation for lesion segmentation in medical images[35,36]. Recently, two groups reported that estimating uncertainty improved the prediction of cancer by digital histopathology[37,38]. However, the uncertainty was estimated for the binary classification. In this study, we have improved the evidential uncertainty and formalized uncertainty thresholding based on the internal validation dataset to conduct confidence evaluation on the testing datasets to detect the fundus anomaly.

In general, compared to these uncertainty approaches, there are advantages of our evidential learning-based uncertainty method: (1) Our UIOS method directly calculates the belief masses of different categories and corresponding uncertainty score by mapping the learned features from the backbone network to the space of Dirichlet parameter distribution. Therefore, our UIOS is trainable end-to-end,

making it easy to implement and deploy; (2) The Dirichlet-based evidential uncertainty method provides well-calibrated uncertainty estimates. It offers reliable uncertainty measurements that align with the true confidence level of the model's predictions, which is supported by the results of this study. This is crucial for applications where accurate assessment of uncertainty is essential, especially for medical diagnosis or critical decision-making scenarios[39,40]. (3) Compared to other uncertainty methods like MC-Drop, ensemble, and TTA, our proposed UIOS can be computationally more efficient. It requires a single forward pass through the model to obtain uncertainty estimates, eliminating the need for multiple model runs or ensemble averaging, thus reducing the computational cost.

In ophthalmology training, junior ophthalmologists usually first learn some common retinal diseases. When they see patients in clinics, they can make diagnosis based on typical manifestations of these common retinal diseases. However, when the disease presentation is not what they have learned, the junior ophthalmologist will feel unconfident in diagnosing the patient and need to consult a senior ophthalmologist. This is a practice to avoid misdiagnosis in clinical practice. Our proposed paradigm in UIOS of uncertainty-inspired open set paradigm mimics the process of reading fundus images by junior ophthalmologists in clinical practice. The proposed uncertainty thresholding strategy enables the model to demand assessment by a human grader, i.e., a senior ophthalmologist, when the model detects high uncertainty in testing OOD samples. It can avoid potential mis-/under-diagnosis incidents in clinical practice and improve the reliability of AI models deployed in clinical practice.

We recognize limitations and the need for improvements in the current study. First, As indicated in Supplementary Table 9, 8.06%, 15.40%, and 30.09% of the samples in the internal testing set and the two external testing sets (TC-JSIEC and TC-unseen) exhibited correct predictions with higher uncertainty than the threshold, resulting in additional labor requirements. Therefore, additional efforts are necessary to enhance the UIOS's ability to learn ambiguous features to further improve its reliability in predicting fundus diseases while reducing the need for manual reconfirmation. Second, we focused solely on classifying fundus images into one main disease category. In the next phase, we will collect more data with multi-label classifications and explore uncertainty evaluation methods for reliable multi-label diseases detection. Third, the model will be tested in more datasets. Samples with high uncertainty scores will be further assessed. Retraining will be performed with the expended dataset. Fourth, our proposed UIOS with the thresholding strategy will be applied to other image modalities (such as OCT, CT, MRI, and histopathology) and combined with artificial intelligence techniques for diagnosing specific diseases.

In conclusion, UIOS model combined with thresholding strategy is capable to accurately classify 9 retinal conditions in the training set and to detect non-target-categories retinal diseases and other OOD samples not seen during training. Our proposed UIOS model can avoid misdiagnoses and provide a robust method for screening retinal anomalies in the real world.

## Methods
### Target categories fundus photo datasets
This study was approved by the Joint Shantou International Eye Center Institutional Review Board and adhered to the principles of the Declaration of Helsinki. The data has been de-identified. In accordance with IRB regulations, if the data does not contain any identifiable patient information, informed consent is not required. As a result, this study has been granted approval to waive the need for informed consent. The clinical assessment and labeling procedure are shown in Supplementary Fig. 4. Fundus images from 5 eye clinics with different models of fundus cameras were collected. Two trained graders performed the annotation independently. If their results were inconsistent, a retinal sub-specialist with more than 10 years

experience would make the final decision. The numbers of images in each category within each dataset are listed in Supplementary Table 15.

We collected 10,034 fundus images of 8 different fundus diseases or normal condition. They were named the primary target-categories (TC) dataset. These images were randomly divided into training (6016), validation (2008) and test sets (2010) in the ratio of 6:2:2. The TC included normal, tigroid fundus (TF), pathological myopia (PM), glaucoma (GL), retinal vein occlusion (RVO), retinal detachment (RD), age-related macular degeneration (AMD), diabetic retinopathy (DR), and central serous chorioretinopathy (CSCR). The inclusion criteria for these diseases are listed in Supplementary Table 16.

There may be several different features in a disease, and different patients may have different features. In human learning, junior doctors usually first learn a few features to begin with and other features later. To investigate the performance of the model in classifying images with different features from the training images, we collected 3,716 fundus images with ambiguous features of the 8 fundus diseases or normal condition as an external testing set (named as unseen target categories, TC-unseen). The including criteria are also listed in Supplementary Table 16.

To further validate the capacity of our proposed UIOS to screen retinal diseases, we also conducted experiments on the public dataset of JSIEC[15], which contained 1000 fundus images from different subjects with 39 types of diseases and conditions. Among them, 435 fundus images were with the target categories and set as the dataset of TC-JSIEC.

## Non-target categories fundus photo datasets
Two non-target categories retinal diseases datasets and one low-quality image dataset were used to investigate the capability of UIOS to detect fundus abnormalities outside the categories of the training set. The first was 1380 fundus images collected from the five clinics with retinal diseases outside the training set as non-target categories (NTC) dataset. The second was 502 images with fundus disease outside the training dataset in the public dataset of JSIEC and set as the dataset of non-target categories from JSIEC1000 (NTC-JSIEC). We removed the images in the categories of massive hard exudate, cotton-wool spots, preretinal hemorrhage, fibrosis and laser spots to avoid confusions caused by multiple morphologic abnormalities. The low-quality dataset was collected from the 5 clinics and consisted of 1066 clinically unusable fundus images due to severe optical opacity, mishandling, or overexposure. The detailed diagnosis of NTC and NTC-JSIEC is listed in Supplementary Table 17.

## Non-fundus photo datasets
Three non-fundus photo public datasets were used to evaluate the performance of AI models in detecting OOD samples. The first was the VOC2012 dataset, with 17,125 natural images of 21 categories[41]. The second was RETOUCH dataset which consisted of 6936 2D retinal optical coherence tomography images[42]. The third was our OCTA dataset collected from our eye clinic, consisting of 304 2D retinal OCTA images.

## Framework of the standard AI model
As shown in Fig. 1, the standard AI model consisted of a backbone network for extracting the feature in formation in fundus images, while a Softmax classifier layer was adopted to produce the prediction results based on the features from the backbone network. For deep learning based disease detections, pre-trained ResNet-50[43] has been widely used as a backbone network to extract the rich feature information contained in medical images and have achieved excellent performance[44–47]. Therefore, in this study, we employed pre-trained ResNet-50 as our backbone network to conduct experiments. As shown in Fig. 1, standard AI model assigned a probability value to each category of fundus diseases that were included in the training set. The

category with the highest probability value was output as the final diagnosis result, without any information reflecting the reliability of the final decision. However, when the standard AI model was given a fundus image of an anomaly out of the fundus diseases in the training set or non-fundus data, the model still output a category of fundus disease from the training set as the final diagnosis result, which could lead to serious mis-/under-diagnosis.

## Framework of UIOS
As shown in Fig. 1, our proposed UIOS architecture was simple and mainly consisted of a backbone network to capture feature information. An uncertainty-based classifier was used to generate the final diagnosis result with an uncertainty score that led to more reliable decision making without losing accuracy. To ensure experimental objectivity, we adopted pre-trained ResNet-50 as our backbone to capture the feature information contained in fundus images. Meanwhile, with fundus images through ResNet-50, the final decision and corresponding overall uncertainty score were obtained by our uncertainty-based classifier, which was mainly composed of three steps. Specifically, this was a $K$-class retinal fundus disease detection.

**Step (1):** Obtaining the evidence feature $E = [e_1, \ldots, e_K]$ for different fundus diseases by applying Softplus activation function to ensure the feature values are larger than 0:

$$E = Softplus(F_{Out}), \tag{1}$$

where $F_{Out}$ was the feature information obtained from the ResNet-50 backbone.

**Step (2):** Parameterizing $E$ to Dirichlet distribution, as:

$$\boldsymbol{\alpha} = E + 1, \text{i.e.}, \alpha_k = e_k + 1, \tag{2}$$

where $\alpha_k$ and $e_k$ are the $k$-th category Dirichlet distribution parameters and evidence, respectively.

**Step (3):** Calculating the belief masses and corresponding uncertainty score as:

$$b_k = \frac{e_k}{S} = \frac{\alpha_k - 1}{S}, u = \frac{K}{S}, \tag{3}$$

where $S = \sum_{k=1}^{K}(e_k + 1) = \sum_{k=1}^{K} \alpha_k$ is the Dirichlet intensities. It can be seen from Eq. 3 the probability assigned to category $k$ is proportional to the observed evidence for category $k$. Conversely, if less total evidence was obtained, the greater the total uncertainty. The belief assignment can be considered as a subjective opinion. The probability of $k$-th retinal fundus disease was computed as $p_k = \frac{\alpha_k}{S}$ based on the Dirichlet distribution[48] (The definition of Dirichlet distribution is detailed in the below section). In addition, to further improve the performance of our UIOS, we also designed a loss function to guide the optimization of our UIOS, the details are shown in section "Loss function."

## Definition of Dirichlet distribution
The Dirichlet distribution was parameterized by its concentration $K$ parameters $\boldsymbol{\alpha} = [\alpha_1, \ldots, \alpha_K]$. Therefore, the probability density function of the Dirichlet distribution was computed as:

$$D(P|\boldsymbol{\alpha}) = \begin{cases} \frac{1}{B(\boldsymbol{\alpha})} \prod_{k=1}^{K} p_k^{\alpha_k - 1} & for\ P \in S_K \\ 0 & Otherwise \end{cases}, \tag{4}$$

where $S_K$ was the $K$-dimensional unit simplex:

$$S_K = \left\{ P | \sum_{k=1}^{K} p_i = 1 \right\}, 0 \le p_i \le 1, \tag{5}$$

and $B(\boldsymbol{\alpha})$ represented the $K$-dimensional multinomial beta function.

## Loss function

Cross entropy loss function has been widely employed in most previous disease detection studies,

$$L_{CE} = -\sum_{k=1}^{K} y_k \log(p_k). \tag{6}$$

In this study, subjective logical (SL) associated the Dirichlet distribution with the belief distribution under the framework of evidential uncertainty theory to obtain the probabilities of different fundus diseases and the corresponding overall uncertainty score based on the evidence collected from the backbone network. Therefore, we could work out the Dirichlet distribution parameter of $\boldsymbol{\alpha} = [\alpha_1, \ldots, \alpha_K]$ and obtained the multinomial opinions $D(p_i | \alpha_i)$, where $p_i$ was the class assignment probabilities on a simplex. Similar to TMC[33], CE loss was modified as:

$$L_{UN} = L_{UN-CE} + \lambda * L_{KL}, \tag{7}$$

where $L_{UN-CE}$ was used to ensure that the correct prediction for each sample yielded more evidence than other classes, while $L_{KL}$ was used to ensure that incorrect predictions would yield less evidence, and $\lambda$ was the balance factor that was gradually increased so as to prevent the model from paying too much attention to the KL divergence in the initial stage of training, which might result in a lack of good exploration of the parameter space and cause the network to output a flat uniform distribution.

$$\begin{aligned} L_{UN-CE} &= \int \left[ \sum_{k=1}^{K} -y_k \log(p_k) \right] \frac{1}{B(\boldsymbol{\alpha}_i)} \prod_{k=1}^{K} p_k^{\alpha_k - 1} dp_k \\ &= \sum_{k=1}^{K} y_k (\psi(S_k) - \psi(\alpha_k)), \end{aligned} \tag{8}$$

where $\psi()$ was the digamma function, while B() is the multinomial beta function for the concentration parameter $\boldsymbol{\alpha}$.

$$L_{KL} = \log \left( \frac{\Gamma\left( \sum_{k=1}^{K} \hat{\alpha}_k \right)}{\Gamma(K) \prod_{k=1}^{K} \Gamma\left( \sum_{k=1}^{K} \hat{\alpha}_k \right)} \right) + \sum_{k=1}^{K} (\hat{\alpha}_k - 1) \left[ \psi(\hat{\alpha}_k) - \psi \sum_{k=1}^{K} \hat{\alpha}_k \right], \tag{9}$$

where $\hat{\boldsymbol{\alpha}} = \boldsymbol{y} + (1 - \boldsymbol{y}) \odot \boldsymbol{\alpha}$ is the adjusted parameter of the Dirichlet distribution which could avoid penalizing the evidence of the ground-truth class to 0, and $\Gamma()$ is the gamma function.

The uncertainty loss $L_{UN}$ could guide the model optimization based on the feature distribution which was parameterized by Dirichlet concentration. However, Dirichlet concentration also changed the original feature distribution, which might cause a decline in the classifier's confidence in the parameterized features. Therefore, to ensure confidence for the parameterized features during training, we further introduced the temperature cross-entropy loss ($L_{TCE}$) to directly guide the model optimization based

on the parameterized features.

$$L_{TCE} = -\sum_{k=1}^{K} y_k \log\left( \frac{b_k}{\tau} \right), \tag{10}$$

where $b_k$ was the belief mass for $k$-th class, while $\tau$ was the temperature coefficients to adjust the belief values distribution, the value is initialized 0.01 was gradually increased to 1 to prevent the low confidence for the belief mass distribution in the initial stage of training.

Therefore, the final loss function for optimizing our proposed model was formalized as (the ablation experiments on the effectiveness of the loss function were shown in Supplementary Table 18):

$$L_{TUN} = L_{UN} + L_{TCE}. \tag{11}$$

## Uncertainty thresholding strategy

In this study, the threshold $\theta$ was determined using the distribution of uncertainty score in our validation dataset. As shown in Supplementary Table 4, the prediction results below the threshold $\theta$ were more likely to be correct, i.e., diagnostic result with high confidence. Conversely, the decisions with an uncertainty score higher than $\theta$ were considered more likely to be unreliable and needed assessment from ophthalmologist. To obtain the optimal threshold value, we calculated the ROC curve, all possible true positive rates (TPRs) and false positive rates (FPRs) for the wrong prediction of validation dataset based on wrong ground truth $\hat{U} = [\hat{u}_1, \ldots, \hat{u}_K]$ and uncertainty scores $U = [u_1, \ldots, u_k]$ for each sample in the validation dataset, n was the total number of samples in the validation dataset, and $\hat{U}$ was obtained by:

$$\hat{u}_i = 1 - \mathbf{1}\{P_i, Y_i\}, where \, \mathbf{1}\{P_i, Y_i\} = \begin{cases} 1 & if \, P_i = Y_i \\ 0 & otherwise \end{cases}, \tag{12}$$

where $P_i$ and $Y_i$ were the final prediction result and ground truth of $i$-th sample in validation dataset. Inspired by Youden's index[49], the objective function based on the TPRs, TPRs, and thresholds of validation dataset was formalized as:

$$\ell(\theta) = 2 * TPRs(\theta) - FPRs(\theta), \tag{13}$$

Therefore, the final optimal threshold value is calculated by $\theta = argmax_\theta \ell(\theta)$. Finally, we obtained the optimal threshold $\theta$ of 0.1158 and the confidence level of a model prediction result:

$$C(P) = \begin{cases} u < \theta, \, high - confidence \\ u \ge \theta, \, low - confidence \end{cases}. \tag{14}$$

## Experimental deployment

We trained our UIOS and other comparison methods including standard AI model, Monte-Carlo drop-out (MC-Drop), ensemble models, time-augmentations (TTA), using entropy across the categorical class probabilities (Entropy), on the public platform Pytorch and Nvidia Geforce RTX 3090 GPU (24 G). Adam was adopted as the optimizer to optimize the model. Its initial learning rate and weight decay were set to 0.0001 and 0.0001, respectively. The batch size was set to 64. To improve the computational efficiency of the model, we resized the image to $256 \times 256$. Meanwhile, online random left-right flipping was applied for data augmentation. In addition, to reduce the time and effort in training multiple models for the ensemble, we used snapshot ensembles[50] to obtain multiple weights for ResNet-50 by using different checkpoints in a single training run. We also compared and analyzed the AUC and F1 scores of different methods.

## Reporting summary

Further information on research design is available in the Nature Portfolio Reporting Summary linked to this article.

## Data availability

Data from JSIEC1000 are available at (https://www.kaggle.com/datasets/linchundan/fundusimage1000). Data from RETOUCH is available at (https://retouch.grand-challenge.org). Data from VOC2012 is available at (http://host.robots.ox.ac.uk/pascal/VOC/voc2012). Additional data sets supporting the findings of this study are not publicly available due to the confidentiality policy of the Chinese National Health Council and institutional patient privacy regulations. However, they are available from the corresponding authors upon request. For replication of the findings and/or further academic and AI-related research activities, data may be requested from corresponding author H.C. (drchenhaoyu@gmail.com), and any requests will be responded to within 10 working days. Source data are provided with this paper.

## Code availability

All codes are available at https://github.com/LooKing9218/UIOS.

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

## Acknowledgements

This research is supported by Agency for Science, Technology and Research (A*STAR) Central Research Fund ("Robust and Trustworthy AI system for Multi-modality Healthcare" to H.F.), Career Development Fund (C222812010 to H.F.), A*STAR Advanced Manufacturing and Engineering (AME) Programmatic Fund (A20H4b0141 to Y.L.), the National Key R&D Program of China (2018 YFA0701700 to H.C. and X.C.), the National Nature Science Foundation of China (U20A20170 to X.C.), Shantou Science and Technology Program (190917085269835 to H.C.), 2020 Li Ka Shing Foundation Cross-Disciplinary Research Grant (2020LKSFG14B to H.C.), the National Natural Science Foundation of China (62136004 to D. Z., 62276130 to D. Z.), and the Key Research and Development Plan of Jiangsu Province (BE2022842 to D. Z.).

## Author contributions

M.W.: conceptualization, methodology, data collection, experimental deployment, software, writing—original draft. T.L.: clinical assessment and annotation and curation, review and editing. L.W.: clinical assessment and curation, experimental deployment, review and editing. A.L: clinical assessment and annotation and curation. K.Z., D.Z., Q.M., C.Z., Y.Q., G.D., Y.Z., Y.P., and W.Z.: methodology, writing—review and editing. X.X., Y.L., and R.S.M.G.: project administration, writing—review. Z.W., J.C., and J.L.: clinical assessment, writing—review and editing. M.Z. and C.P.P.: clinical assessment, writing—review and editing. H.C.: supervision, clinical assessment and annotation and curation, clinical support, writing—review and editing. X.C. and H.F.: supervision, project administration, methodology, writing—review and editing.

## Competing interests

The authors declare no competing interests.
