## [Peer Review File · Nature Communications]

Uncertainty-inspired Open Set Learning for Retinal Anomaly IdentificationREVIEWER COMMENTS

Reviewer #1 (Remarks to the Author):

In this study, Wang et al develop a novel method of "Uncertainty-inspired Open Set Learning" (UIOS) to potentially improve the clinical deployment of AI models examining for retinal abnormalities from retinal photographs. They trained their AI models in the identification of 9 different retinal conditions, but for their UIOS model also built in an "uncertainty score" that helps to flag up images where the predictions are less certain, and which may benefit from a human grader evaluation. This also has the advantage of avoiding a straight misclassification when the model encounters conditions different from the original training dataset, or in cases of low quality or data outside of distribution. They compare this UIOS model against a "standard AI model" in a few datasets, and show that their UIOS model demonstrates superior performance.

I think that this is an important step towards improving clinical deployment of AI models for medical image classification, and in theory, helps to bridge an important gap between research and clinical translation. It is likely not going to be feasible to train AI models on such large and diverse datasets, that they will be able to handle the entire wide range of abnormalities and challenges that arise in images from clinical practice. Hence, this "uncertainty" based approach makes sense, and is very appealing in this regard.

However, this does not mean that this is the only way towards clinical translation and deployment of AI models. There are already autonomous AI models for diabetic retinopathy (DR) screening that have received regulatory approval and are in clinical use. They do not rely on "uncertainty scores", and instead their approach is usually to perform an image quality assessment, screen out poor quality images, and then only predict the presence/absence of that particular disease that the AI model is trained for. These models come with disclaimers that they do not provide assessments of any disease other than DR for example. This is in practice also a workable approach. Nevertheless, this study by Wang et al takes a different approach, which has potential.

I have the following concerns about the manuscript:

1. I understand how the uncertainty prediction is useful when the algorithm encounters atypical examples of the conditions it has been previously trained on, or when it encounters diseases it has not seen before. That would explain why the UIOS model outperforms the standard AI model. However, I also note that on average, even in the internal testing datasets, and on the CJSIEC dataset, where the algorithm sees the same diseases it has been trained on, the UIOS already outperforms the standard AI model. Is there a reason why this might be the case? It begs the question as to whether the comparison is fair – is the standard AI model a fair comparator arm? Other than the uncertainty prediction, are there other technical differences between the UIOS and the standard AI model that would account for these performance differences? I don't see why the UIOS should outperform the standard AI model, on diseases that were already in the training dataset, and where the images are "typical".

2. While I understand the authors' intentions to investigate a group of retinal diseases within their original training dataset, as well as a group of other retinal diseases that the algorithm was not initially trained on, I think that terminology and naming of the datasets is an issue. The authors list a number of "Common Retinal Diseases", and other diseases outside this list are considered "Rare Retinal Diseases", but these terms are inaccurate. For example, epiretinal membranes are considered "Rare", while retinal detachments are considered "Common", but the prevalence of epiretinal membranes is considerably higher than that of retinal detachments. I would consider renaming the datasets in the interest of precision and clarity.

3. On a similar note, labelling the "Common" conditions as "Typical" and "Non-typical" is also

potentially confusing to a clinical audience, based on the definitions provided in Supplementary Table 6. The clinical definitions provided there are not necessarily “typical” or “non-typical”. For example in DR, “typical” DR is defined as “multiple microaneurysms, variable dot/blot-like hemorrhages, hard exudates, maybe with macular edema, neovascularization, vitreous/preretinal hemorrhage or preretinal proliferative membrane”, while “non-typical” DR is defined as “only microaneurysms (mild NPDR) or severe proliferative membrane and vitreous hemorrhage covering the retinal characteristics. Any stages with laser spots”. Mild NPDR is probably the most common manifestation among eyes with DR, and cannot be considered “non-typical” DR. The authors clearly intend to define this category as “different” or “outside the original training set”, rather than “non-typical”, and so terminology here needs to be improved to avoid confusion and inaccuracy.

4. Especially in the RJSIEC dataset, many of the image labels are likely to overlap in images, as some of the labels are diagnoses, but others are just morphologic abnormalities, and it is likely that many of these abnormalities would co-exist, and that images would qualify for multiple different labels. It is not clear if these images had multiple labels with overlap, or if they each were assigned only one label – which would be inaccurate.

5. I am unclear about the difference between the UIOS model and the UIOS model + thresholding. I would think that the UIOS model would originally have to have a threshold value already determined from the distribution of uncertainty values in the validation dataset. Without any existing threshold, there would be no use to the model. So does that mean that in the “UIOS model + thresholding”, for each task and dataset that the UIOS model is being applied to, a new “optimum threshold” is being re-selected? If the model requires new or different thresholds each time for different datasets and settings, then that limits its generalizability, and it needs to be made very clear in the results, that these are obtained with custom or “optimized” thresholds specific to each dataset and task. This must be more clearly emphasized in the paper.

6. What happens when an image is flagged up as “uncertain” beyond the uncertainty threshold, and the image needs to be reviewed by an ophthalmologist? In real world deployment, the patient would be sent to an ophthalmologist for review, but in this study, what happens to that image or eye in terms of calculating the diagnostic performance or F1? Is that eye removed from analysis? Or does the image get graded by an ophthalmologist, and then the grading from that ophthalmologist go into the diagnostic performance results? That would mean that the UIOS model achieves better results, but is of course less automated, and is a more resource-intensive approach, which needs to be clearly acknowledged.

7. Related to this, another important metric to consider and present clearly, is in the “typical” images of “common retinal disease”, how often did the UIOS model unnecessarily flag the images as uncertain and prompt a human grading? For example, in these typical common images, how often did the standard AI model and the UIOS model both make the correct disease prediction, but the uncertainty value was beyond threshold? That is an important metric to consider in terms of the potential disadvantages of the UIOS model.

8. The performance of the standard AI model on the rare disease datasets and the ODD datasets should also be reported, even though we expect this to be very poor.

9. In general, I think that there needs to be much more robust discussion of the limitations of the UIOS model and approach.

10. English language editing is required for this paper

Reviewer #2 (Remarks to the Author):

This work addresses the problem of open set recognition in the context of the detection of 8 disease categories (9 classes=8 disease+1 healthy class) in retinal fundus images. Although AI models can only predict a limited set of categories available during training, once it is deployed in the real-world, the model can encounter out-of-distribution data (OOD). Thus, a useful AI model should not only be able to predict the class probabilities but also be able to quantify the uncertainty in its predictions.

As noted in the manuscript, different methods exist to predict uncertainty in Deep learning. For example, Bayesian Neural Networks [18, 23-25] and its more scalable approximation with Monte Carlo Dropout [26], construction of ensembles to measure the variance in multiple predictions either by training several models [27, 28] or applying different test time augmentations [29], and evidential deep learning which aims to directly learn the parameters of the distribution over different model weights [30-32]. This work seems to use the method in [31] for the uncertainty estimation.

The OOD data could be of various types: (a) the image belongs to one of the classes known to the AI model, but the features determining the class is ambiguous due to shift in population distribution or scanner type (eg., CJSIEC dataset); (b) ambiguity in the biomarkers that can be seen in the fundus images (Non Typical CRD dataset); (c) images having other retinal diseases not encountered during training (RJSIEC and RRD datasets); and (d) ability to detect non-fundus images which may be input in the real-world by mistake (eg., natural images from VOC-12 or other retinal imaging modalities such as OCT or OCT-A). This work has exhaustively shown good performance on each of these different scenarios. However, it has not been benchmarked against other uncertainty estimation methods.

Major Comments:

I. This work employs an existing work [31] on evidential deep learning and applies it to retinal imaging. I did not find any novel technical contribution in terms of the methodology. The authors should explicitly clarify if they have made any new technical contributions and how is the proposed method different from that of [31].

II. The proposed method (based on [31]) has not been benchmarked against other existing methods for uncertainty estimation especially, (1) Monte-Carlo drop-out, (2) ensemble models and (3) application of different test time-augmentations and (4) using entropy across the categorical class probabilities in the standard AI model also gives an estimate of uncertainty. To reduce the time and effort in training multiple models for the ensemble, the authors can use snapshot ensembling (Huang et. al. "Snapshot Ensembles: Train 1, Get M for Free", ICLR 2017) to obtain multiple weights for ResNet-50 by using different checkpoints in a single training run. The authors should benchmark these methods both on Internal test set, CJSIEC set and non-typical CRD set as in Fig. 2 and also on the anomaly detection task on the RRD, RJSIEC, low-quality, VOC, OCTA and RETOUCH datasets to indicate what percentage of the images could it detect as OOD? Does the evidential learning based method have a better performance compared to the other methods?

III. The performance in terms of AUROC as shown in Fig. 2. In terms of the average AUROC across all classes, the performance difference between the standard AI model, UIOS and UIOS+threshold is very small for the internal test set (0.9968 vs 0.9979 vs 0.9989) and CJSIEC set (0.9767 vs 0.9907 vs 0.9977). A statistical significance test based on the DeLong's test needs to be performed to show that the difference in performance is statistically significant. Similar tests with p-values should also be reported for the benchmark methods (see point II above).

Minor Comments:

IV. In terms of implementation details in Sec 3.9

a. Why is only random flipping used for data augmentation? Typically small rotation about the image center, color jitter, random resize-crop are also used while training deep learning models to improve generalization.

b) Is the GPU mentioned DGX 3090 correct or should it be Nvidia Geforce RTX 3090 ? I wanted to confirm as I have not heard of the DGX 3090 GPU before.

Response to Reviewers' Comments

Paper ID: NCOMMS-23-15702-T

We thank the associate editor and the reviewers for their excellent comments and suggestions which have helped us to greatly improve the quality of the paper. We have addressed all issues raised by the editor and reviewers and have revised the manuscript accordingly. The changes are marked in red font in the resubmitted version to facilitate the review process. In the response texts below, all references to figures, pages and citations correspond to the resubmitted version.

Reviewers #1 (Remarks to the Author):

1. In this study, Wang et al develop a novel method of “Uncertainty-inspired Open Set Learning” (UIOS) to potentially improve the clinical deployment of AI models examining for retinal abnormalities from retinal photographs. They trained their AI models in the identification of 9 different retinal conditions, but for their UIOS model also built in an “uncertainty score” that helps to flag up images where the predictions are less certain, and which may benefit from a human grader evaluation. This also has the advantage of avoiding a straight misclassification when the model encounters conditions different from the original training dataset, or in cases of low quality or data outside of distribution. They compare this UIOS model against a “standard AI model” in a few datasets, and show that their UIOS model demonstrates superior performance. I think that this is an important step towards improving clinical deployment of AI models for medical image classification, and in theory, helps to bridge an important gap between research and clinical translation. It is likely not going to be feasible to train AI models on such large and diverse datasets, that they will be able to handle the entire wide range of abnormalities and challenges that arise in images from clinical practice. Hence, this “uncertainty” based approach makes sense, and is very appealing in this regard. However, this does not mean that this is the only way towards clinical translation and deployment of AI models. There are already autonomous AI models for diabetic retinopathy (DR) screening that have received regulatory approval and are in clinical use. They do not rely on “uncertainty scores”, and instead their approach is usually to perform an image quality assessment, screen out poor quality images, and then only predict the presence/absence of that particular disease that the AI model is trained for. These models come with disclaimers that they do not provide assessments of any disease other than DR for example. This is in practice also a workable approach. Nevertheless, this study by Wang et al takes a different approach, which has potential.

REPLY: We would like to thank you for your professional review work, constructive comments, and valuable suggestions on our manuscript. As you are concerned, there are several issues need to be addressed, which are replied in detail as below.

2. I understand how the uncertainty prediction is useful when the algorithm encounters atypical examples of the conditions it has been previously trained on, or when it encounters diseases it has not seen before. That would explain why the UIOS model outperforms the standard AI model. However, I also note that on average, even in the internal testing datasets, and on the CJSIEC dataset, where the algorithm sees the same diseases it has been trained on, the UIOS already outperforms the standard AI model. Is there a reason why this might be the case? It begs the question as to whether the comparison is fair – is the standard AI model a fair comparator arm? Other than the uncertainty prediction, are there other technical differences between the UIOS and the standard AI model that

would account for these performance differences? I don't see why the UIOS should outperform the standard AI model, on diseases that were already in the training dataset, and where the images are "typical".

REPLY: Thanks for your question. In this study, although both the proposed UIOS model and the standard AI model were based on the pre-trained ResNet50 to obtain feature information, different classifiers were used to predict the results. The standard AI model used the Softmax classifier commonly used in classification tasks to directly output the prediction results, while our UIOS model utilized an uncertainty-based classifier, which mapped the feature information obtained from the pre-trained ResNet50 to the Dirichlet distribution to reach the final decision with the corresponding uncertainty score. And, to optimize our UIOS model, we introduced a series of loss functions based on the Dirichlet distribution, which utilized uncertainty constraints to guide the optimization process. This approach enhances the stability of the model, leading to more reliable prediction results. Therefore, the two models were trained based on different classifier using different loss function, thus leading to different performance of the two models on the same dataset. More details about the framework of standard AI model and our UIOS model are given in Sec. 3.4 and Sec. 3.5, respectively, and the loss function was detailed in Sec. 3.7.

3. While I understand the authors' intentions to investigate a group of retinal diseases within their original training dataset, as well as a group of other retinal diseases that the algorithm was not initially trained on, I think that terminology and naming of the datasets is an issue. The authors list a number of "Common Retinal Diseases", and other diseases outside this list are considered "Rare Retinal Diseases", but these terms are inaccurate. For example, epiretinal membranes are considered "Rare", while retinal detachments are considered "Common", but the prevalence of epiretinal membranes is considerably higher than that of retinal detachments. I would consider renaming the datasets in the interest of precision and clarity.

REPLY: Thanks for your professional suggestions. We are sorry for the confusions about the "Common" and "Rare" definitions caused by inappropriate terminology. Our current data classification was based on the amount of collected fundus photography. In this study, 9 conditions of relatively large sample sizes were categorized as the target diseases for detection, which had been defined as "Common" for training, while 6 conditions that we had less data were defined as "Rare" for testing. However, as you have rightly pointed out, the definitions of "Common" and "Rare" are not epidemiologically correct. So, we replaced "Common" with "Target" and "Rare" with "Non-target" in the revised manuscript, dividing the 15 conditions into 9 Target Categories (TC) and 6 Non-target Categories (NTC). More details are given in Sec. 3.1-3.3, Supplementary Table 6 and Supplementary Table 7.

4. On a similar note, labelling the "Common" conditions as "Typical" and "Nontypical" is also potentially confusing to a clinical audience, based on the definitions provided in Supplementary Table 6. The clinical definitions provided there are not necessarily "typical" or "non-typical". For example in DR, "typical" DR is defined as "multiple microaneurysms, variable dot/blot-like hemorrhages, hard exudates, maybe with macular edema, neovascularization, vitreous/preretinal hemorrhage or preretinal proliferative membrane", while "non-typical" DR is defined as "only microaneurysms (mild NPDR) or severe proliferative membrane and vitreous hemorrhage covering the retinal characteristics. Any stages with laser spots". Mild NPDR is the probably the most common manifestation among eyes with DR, and cannot be considered "non-typical" DR. The authors clearly intend to define this category as "different" or "outside the original training set", rather than

“nontypical”, and so terminology here needs to be improved to avoid confusion and inaccuracy.

REPLY: Thanks for your kind reminder. Sorry for the confusion caused by the terms “Typical” and “non-typical”. In this study, we intentionally defined representative lesions as “typical” for training, while indistinct features as “non-typical” for testing. Unfortunately, such terms were inappropriate and could confuse clinicians. Thus, in the revised manuscript, we replaced “Typical” with “Seen” to indicate that the features in training set were supposed to be learned by the model. We also replaced “Non-typical” with “Unseen” to show the remaining features as not yet learned by our model, which would be used to test the uncertainty. More details are given in Sec. 3.1-3.3, Supplementary Table 6 and Supplementary Table 7.

5. Especially in the RJSIEC dataset, many of the image labels are likely to overlap in images, as some of the labels are diagnoses, but others are just morphologic abnormalities, and it is likely that many of these abnormalities would co-exist, and that images would qualify for multiple different labels. It is not clear if these images had multiple labels with overlap, or if they each were assigned only one label –which would be inaccurate.

REPLY: Thank you very much for raising this important point. The non-target categories from JSIEC1000(NTC-JSIEC) were divided from the JSIEC-1000 dataset, which is a public dataset with 1000 fundus images of 39 categories. All the images were assigned only one label, not multiple labels. The categories included both diagnosis and morphologic abnormalities. The classification was based on the major feature of the images. The characters in different categories were different. However, to avoid confusion, we removed the images in the categories of massive hard exudate, cotton-wool spots, preretinal hemorrhage, fibrosis and laser spots (As shown in Sec. 3.2, Supplementary Table 6, and Supplementary Table 7). The results were analyzed and revised (As shown in Sec. 1.3) in the revised manuscript. We have also added some discussion that the current model is for multi-category, and further work for multi-label will be conducted (Sec. 2, row 5-16 in Page 10).

6. I am unclear about the difference between the UIOS model and the UIOS model + thresholding. I would think that the UIOS model would originally have to have a threshold value already determined from the distribution of uncertainty values in the validation dataset. Without any existing threshold, there would be no use to the model. So does that mean that in the “UIOS model + thresholding”, for each task and dataset that the UIOS model is being applied to, a new “optimum threshold” is being re-selected? If the model requires new or different thresholds each time for different datasets and settings, then that limits its generalizability, and it needs to be made very clear in the results, that these are obtained with custom or “optimized” thresholds specific to each dataset and task. This must be more clearly emphasized in the paper.

REPLY: Thanks for your suggestions. In the current study, the threshold was determined using the distribution of uncertainty score in the validation dataset. All datasets and tasks used this threshold. For further clinical implementation, the optimal threshold in our task can be used directly (Detailed in section 3.8 Uncertainty thresholding strategy).

7. What happens when an image is flagged up as “uncertain” beyond the uncertainty threshold, and the image needs to be reviewed by an ophthalmologist? In real world deployment, the patient would be sent to an ophthalmologist for review, but in this study, what happens to that image or eye in terms of calculating the diagnostic performance or F1? Is that eye removed from analysis? Or does the

image get graded by an ophthalmologist, and then the grading from that ophthalmologist go into the diagnostic performance results? That would mean that the UIOS model achieves better results, but is of course less automated, and is a more resource-intensive approach, which needs to be clearly acknowledged.

REPLY: Thank you for the very helpful comments. Yes, the images flagged up as “uncertain” beyond the uncertainty threshold needed to be re-evaluated by ophthalmologists. These images were removed in calculating the diagnostic performance metrics. We have clarified this point in the revised manuscript (Sec. 2, Page 9-10).

We noted that while the UIOS model may be resource-intensive by suggesting the samples with high uncertainty scores above the threshold value to seek double-checking from their ophthalmologist, it is still more intelligent and better aligned with clinical reality demands than the standard AI model. To date, although many AI-based algorithms and products have been developed and applied in some clinical practices, AI remains as an auxiliary tool to assist doctors in disease screening and evaluation. One reason is the lack of corresponding confidence assessment for AI predictions. However, standard AI model directly provides the final output as the final diagnosis result without giving any reference for confidence assessment of the final prediction. As a result, doctors do not know which samples’ predictions are reliable, leading to two probable extreme scenarios: all samples with AI predictions are sought for doctors’ reconfirmation to avoid mis-/under diagnosis issues, which greatly diminishes the benefits of deploying AI algorithms to reduce workload of busy doctors in clinical practice. Conversely, there is complete acceptance of AI model predictions that can result in serious medical incidents due to the mis-/under-diagnosis. In contrast to standard AI model, our UIOS model output the final decision with corresponding uncertainty score to indicate the reliability of the prediction. By setting a threshold value, it can explicitly remind ophthalmologists whether the prediction result is reliable or not. Therefore, compared to the standard AI model, our UIOS model is more intelligent and better aligned with the demands of real world clinical practice than standard AI model.

Meanwhile, following your suggestions, we added in the revised manuscript comprehensive analysis of the disadvantages and advantages of our UIOS.

8. Related to this, another important metric to consider and present clearly, is in the “typical” images of “common retinal disease”, how often did the UIOS model unnecessarily flag the images as uncertain and prompt a human grading? For example, in these typical common images, how often did the standard AI model and the UIOS model both make the correct disease prediction, but the uncertainty value was beyond threshold? That is an important metric to consider in terms of the potential disadvantages of the UIOS model.

REPLY: Thanks again for your helpful suggestions. As shown in Supplementary Table 9, 9.75% of the images in the “Typical Common retinal disease” (Or Target-seen Categories as mentioned above) internal testing set were above uncertainty threshold and flagged up for double-check. We have noted that standard AI model could only predict the final prediction result without uncertainty score to evaluate the reliability of the prediction result. Therefore, to address this concern, we have assessed how often the proposed UIOS made the correct disease prediction, but the uncertainty value was beyond threshold. As shown in Supplementary Table 18, 8.06%, 15.40%, and 30.99% samples obtained correct predictions with higher uncertainty than threshold in internal testing set and two external testing sets of target categories from JSIEC1000 (TC-JSIEC) and unseen target categories (TC-unseen) dataset, respectively. Therefore, while our proposed UIOS has performed well in enhancing the reliability of fundus disease detection in fundus images and screening for out-of-distribution (OOD) abnormalities, further efforts are needed to enhance the capability to learn ambiguous features. We should improve UIOS’s reliability in predicting fundus diseases while reducing the necessity for manual

reconfirmation. This need has been added in the Discussion section of the revised manuscript.

Supplementary Table 9: Rates for prompting a human grading and correct disease prediction with high uncertainty above threshold

Category	Internal testing set (%)	TC-JSIEC (%)	TC-unseen dataset (%)
Rate for prompting a human grading	9.75	23.22	47.55
Rate of correct disease prediction with high uncertainty above threshold	8.06	15.40	30.09

9. The performance of the standard AI model on the rare disease datasets and the OOD datasets should also be reported, even though we expect this to be very poor.

REPLY: Thank you for the suggestion. Since all samples in the rare disease datasets and the OOD datasets were outside the 9 categories contained in the training dataset, and the standard AI model could only predict the 9 diseases contained in the training categories, predictions of the samples in the rare disease datasets and the OOD datasets were more likely to be incorrect. In addition, in Figure 5, we also demonstrated the prediction examples of standard AI model on rare disease datasets and the OOD datasets, which consistently yielded erroneous predictions (As shown in Sec. 1.3).

10. In general, I think that there needs to be much more robust discussion of the limitations of the UIOS model and approach.

REPLY: Thanks for another excellent suggestion. We have added more discussion about the limitations of the UIOS model in the revised manuscript.

We recognize some limitations and the need for improvements in the current study. First, As indicated in Supplementary Table 9, 8.06%, 15.40%, and 30.09% of the samples in the internal testing set and the two external testing sets (TC-JSIEC and TC-unseen) exhibited correct predictions with higher uncertainty than the threshold, resulting in additional labor requirements. Therefore, additional efforts are necessary to enhance the UIOS's ability to learn ambiguous features to further improve its reliability in predicting fundus diseases while reducing the need for manual reconfirmation. Second, we focused solely on classifying fundus images into one main disease category. In the next phase, we shall collect more data with multi-label classifications and explore uncertainty evaluation methods for reliable multi-label diseases detection. Third, the model will be tested in more datasets. Samples with high uncertainty scores will be further assessed. Retraining will be performed with the expended dataset. Fourth, our proposed UIOS with the thresholding strategy will be applied to other image modalities (such as OCT, CT, MRI, and histopathology) and combined with artificial intelligence techniques for diagnosing specific diseases.

11. English language editing is required for this paper.

REPLY: We sincerely thanks you for your feedback which would help to improve the quality of our manuscript. We have proofed and correct English language editing in the revised manuscript, and we hope the revised manuscript could be acceptable.

Reviewer #2

Comments to the Author

This work addresses the problem of open set recognition in the context of the detection of 8 disease categories (9 classes=8 disease+1 healthy class) in retinal fundus images. Although AI models can only predict a limited set of categories available during training, once it is deployed in the real-world, the model can encounter out-of-distribution data (OOD). Thus, a useful AI model should not only be able to predict the class probabilities but also be able to quantify the uncertainty in its predictions.

As noted in the manuscript, different methods exist to predict uncertainty in Deep learning. For example, Bayesian Neural Networks [18, 23-25] and its more scalable approximation with Monte Carlo Dropout [26], construction of ensembles to measure the variance in multiple predictions either by training several models [27, 28] or applying different test time augmentations [29], and evidential deep learning which aims to directly learn the parameters of the distribution over different model weights [30-32]. This work seems to use the method in [31] for the uncertainty estimation.

The OOD data could be of various types: (a) the image belongs to one of the classes known to the AI model, but the features determining the class is ambiguous due to shift in population distribution or scanner type (eg., CJSIEC dataset); (b) ambiguity in the biomarkers that can be seen in the fundus images (Non Typical CRD dataset); (c) images having other retinal diseases not encountered during training (RJSIEC and RRD datasets); and (d) ability to detect non-fundus images which may be input in the real-world by mistake (eg., natural images from VOC-12 or other retinal imaging modalities such as OCT or OCT-A). This work has exhaustively shown good performance on each of these different scenarios. However, it has not been benchmarked against other uncertainty estimation methods.

REPLY: We would like to thank you for your professional review work, constructive comments, and valuable suggestions on our manuscript. Your time and efforts are greatly appreciated. We have strived to improve the manuscript accordingly as listed in details below.

Major Comments:

1. This work employs an existing work [31] on evidential deep learning and applies it to retinal imaging. I did not find any novel technical contribution in terms of the methodology. The authors should explicitly clarify if they have made any new technical contributions and how is the proposed method different from that of [31].

REPLY: Thanks for your comments. In the revised manuscript, the index of conference [31] has been updated to [33], compared to [33], we have two main contributions:

1) We improved the loss function, which can guide the UIOS to learn the uncertainty distribution to assess the reliability of the decision while improving the confidence in the belief mass, so that UIOS can avoid performance degradation while giving reliability predictions. In Supplementary Table 8, we also conducted experiments to verify the effectiveness of UIOS compared with the method proposed in [33] (Backbone+ L_{UN}). As shown in Supplementary Table 18, the average F1 score of [33] (Backbone+ L_{UN}) on most testing sets was lower than that of Backbone, mainly because Dirichlet re-parameterization changed the original feature distribution that reduced the model's confidence in the class-related evidence and led to lower performance. Focusing on this issue, we improved the loss function by introducing a temperature cross-entropy loss function, which could enhance UIOS's confidence in the features that were re-parameterized by Dirichlet, thereby improving the performance in detecting retinal fundus diseases. Consequently, as presented in Supplementary Table 8, our proposed UIOS has achieved the highest performance compared to Backbone and [33] (Backbone+ L_{UN}) on the internal testing set, and two external test sets, the TC-JSIEC dataset and TC-unseen

dataset, both of which have different feature distributions from the training data. Compared to [33] (Backbone+ L_{UN}), UIOS has achieved an F1 score improvement of 3.48%, 14.64%, and 19.21% on three test sets, reaching 97.29%, 87.19%, and 77.15%, respectively. These experimental results further demonstrate the effectiveness of the proposed UIOS, which improves the reliability of prediction with higher performance (Supplementary Table 8).

2) We developed a thresholding strategy to explicitly select the optimal uncertainty threshold value θ to quantify the confidence for the final decision. The prediction results below the threshold θ were more likely to be correct, i.e., diagnostic result with high confidence. Conversely, the decisions with an uncertainty score higher than θ were considered more likely to be unreliable and needed assessment from ophthalmologist (Sec. 3.8).

2. The proposed method (based on [31]) has not been benchmarked against other existing methods for uncertainty estimation especially, (1) Monte-Carlo drop-out, (2) ensemble models and (3) application of different test time-augmentations and (4) using entropy across the categorical class probabilities in the standard AI model also gives an estimate of uncertainty. To reduce the time and effort in training multiple models for the ensemble, the authors can use snapshot ensembling (Huang et al. "Snapshot Ensembles: Train 1, Get M for Free", ICLR 2017) to obtain multiple weights for ResNet-50 by using different checkpoints in a single training run. The authors should benchmark these methods both on Internal test set, CJSIEC set and non-typical CRD set as in Fig. 2 and also on the anomaly detection task on the RRD, RJSIEC, low-quality, VOC, OCTA and RETOUCH datasets to indicate what percentage of the images could it detect as OOD? Does the evidential learning-based method have a better performance compared to the other methods?

REPLY: Thanks for the helpful comments. We have conducted more experiments to comprehensively analyze the performance of different uncertainty-based methods. The results revealed better performance of UIOS than other uncertainty-based methods. In the revised manuscript, we summarized three advantages of our evidential learning-based uncertainty method compared to other uncertainty approaches in the Discussion section: 1. Our UIOS method directly calculated the belief masses of different categories and corresponding uncertainty score by mapping the learned features from the backbone network to the space of Dirichlet parameter distribution. Therefore, our UIOS is trainable end-to-end, which transcribes to easy implementation and deployment; 2. The Dirichlet-based evidential uncertainty method provides well-calibrated uncertainty estimates. It offers reliable uncertainty measurements that align with the true confidence level of UIOS's predictions. This is crucial for applications where accurate assessment of uncertainty is essential, especially for medical diagnosis or critical decision-making scenarios [39-40]. 3. Compared to some other uncertainty methods like MC-Drop, ensemble, and TTA, our proposed UIOS can be computationally more efficient. It requires a single forward pass through the model to obtain uncertainty estimates, eliminating the need for multiple models runs or ensemble averaging, thus reducing the computational cost.

[39] Gawlikowski J, Tassi C R N, Ali M, et al. A survey of uncertainty in deep neural networks. arXiv preprint arXiv:2107.03342, 2021.

[40] Abdar M, Pourpanah F, Hussain S, et al. A review of uncertainty quantification in deep learning: Techniques, applications and challenges. Information Fusion, 2021, 76: 243-297.

Compared to other baseline methods such as Monte-Carlo drop-out (MC-Drop), Ensemble, test time-

augmentation (TTA), and entropy, UIOS achieved the highest average F1 score on internal testing set, and two external dataset of TC-JSIEC and TC-unseen dataset (Supplementary Table 5~6 and 10~13). Meanwhile, the F1-scores of all methods have been further improved after assigning the samples with high uncertainty score above the threshold value to seek double-checking from ophthalmologists. Notably, UIOS still obtained higher average F1 scores than other baseline approaches. These experimental results demonstrate higher prediction precision by UIOS for samples considered to have higher confidence after screening by the uncertainty threshold value, which is more in line with real world disease screening by junior doctors in clinical practice and effectively improves model credibility and robustness. The AUCs of different methods have been added in the revised manuscript. UIOS achieved higher AUC than other uncertainty-based methods on most datasets (Supplementary Fig. 2). In addition, although the average AUC of UIOS+thresholding was slightly lower than MC-Drop+thresholding and Entropy+thresholding on the internal test dataset, it still achieved a higher average AUROC on most of the datasets (Supplementary Fig. 3). In further experiments to verify the abnormal detection performance of different methods based on the uncertainty thresholding strategy, UIOS model outperformed other baseline methods significantly on most datasets, further affirming its capability to screen for rare fundus diseases, low-quality data, and non-fundus OOD samples (Supplementary Table 14).

Supplementary Table 5: F1 scores of different methods on internal testing set

Category	MC-Drop	Ensemble	TTA	Entropy	UIOS
Normal	98.82	97.69	95.27	97.67	99.18
TF	96.45	94.18	94.74	91.30	93.12
PM	91.54	97.66	97.94	95.18	98.84
GL	91.15	93.58	95.49	93.09	98.53
RVO	93.58	96.44	95.49	96.12	97.36
RD	95.99	96.38	96.74	92.13	99.27
AMD	91.01	90.98	92.56	85.60	97.24
DR	94.59	95.41	94.79	91.04	98.04
CSCR	89.02	89.02	83.23	81.03	94.05
Average	93.57	94.59	94.03	91.46	97.29

Supplementary Table 6: F1 scores of different methods on internal testing set after thresholding

Category	MC-Drop	Ensemble	TTA	Entropy	UIOS
Normal	99.87	99.37	97.97	99.88	99.88
TF	97.56	98.59	98.09	98.99	98.68
PM	97.44	99.66	99.00	100.00	99.39
GL	97.71	99.49	98.48	99.73	100.00
RVO	99.07	98.36	98.02	99.53	99.60
RD	99.40	99.18	99.22	99.38	100.00
AMD	97.23	98.98	96.14	98.54	99.41
DR	98.84	98.87	98.21	99.44	99.62
CSCR	98.97	95.45	86.54	96.00	99.33
Average	98.45	98.66	96.85	99.05	99.55

Supplementary Table 10: F1 scores of different methods on TC-JSIEC set

Category	MC-Drop	Ensemble	TTA	Entropy	UIOS
Normal	27.27	68.42	71.60	58.06	84.34
TF	70.59	81.25	86.67	85.71	78.79
PM	97.20	99.08	95.41	94.64	100.00
GL	66.67	80.00	66.67	56.00	72.73
RVO	87.18	89.08	88.00	90.32	95.24
RD	92.45	96.36	93.46	91.43	94.44
AMD	75.13	93.08	90.00	88.05	93.67
DR	87.83	88.66	87.50	86.91	87.76
CSCR	56.52	70.00	71.79	48.28	77.78
Average	73.43	85.10	83.46	77.71	87.19

Supplementary Table 11: F1 scores of different methods on TC-unseen dataset

Category	MC-Drop	Ensemble	TTA	Entropy	UIOS
Normal	66.28	72.54	69.80	74.21	83.17
TF	54.82	49.94	60.53	54.17	78.43
PM	43.99	72.22	75.43	65.38	80.00
GL	68.11	73.00	74.23	74.37	78.33
RVO	67.57	60.23	74.09	65.21	84.96
RD	61.20	60.68	57.95	61.84	72.19
AMD	46.83	44.05	41.20	39.14	50.17
DR	72.04	69.68	69.67	79.47	83.21
CSCR	73.48	74.53	69.96	74.84	83.84
Average	61.59	64.10	65.87	65.40	77.15

Supplementary Table 12: F1 scores of different methods on TC-JSIEC set after thresholding

Category	MC-Drop	Ensemble	TTA	Entropy	UIOS
Normal	22.22	66.67	71.19	40.00	90.00
TF	92.31	100.00	95.24	50.00	94.74
PM	100.00	100.00	99.01	99.05	100.00
GL	90.91	100.00	76.92	66.67	93.33
RVO	91.30	91.18	94.64	95.83	100.00
RD	100.00	100.00	98.90	98.36	98.85
AMD	85.31	96.97	96.30	95.74	99.31
DR	89.71	98.63	92.57	98.46	96.83
CSCR	76.19	85.71	78.79	88.00	100.00
Average	83.11	93.24	89.28	81.35	97.01

Supplementary Table 13: F1 scores of different methods on TC-unseen dataset after thresholding

Category	MC-Drop	Ensemble	TTA	Entropy	UIOS
Normal	82.89	78.15	79.86	87.68	92.86
TF	72.89	55.59	64.40	55.46	89.14
PM	45.28	84.54	84.18	88.24	94.69
GL	84.40	92.60	83.10	94.31	95.08
RVO	74.56	63.89	81.05	80.19	97.03
RD	81.40	84.48	68.33	85.11	92.59
AMD	58.71	73.91	54.66	70.00	76.63
DR	81.91	80.31	75.83	93.46	96.04
CSCR	87.92	89.81	74.87	89.49	93.12
Average	74.44	78.14	74.03	82.66	91.91

Supplementary Table 14: The abnormal detection rates of different methods on different datasets

Methods	NTC	NTC-JSIEC	Low-quality	RETOUCH	OCTA	VOC 2012	Time (ms/per image)
MC-Drop	70.80	71.71	54.13	1.38	54.28	73.82	18.11
Ensemble	79.20	81.08	71.11	5.49	100.00	83.85	1.01
CE	82.61	83.27	84.62	2.72	0.00	44.85	0.34
TTA	58.83	55.98	53.85	26.92	5.59	48.87	4.87
UIOS	86.67	82.27	89.40	99.81	99.01	96.18	0.34

Supplementary Fig. 2. The receiver operating characteristic (ROC) curves of our UIOS model and other uncertainty-based methods in internal and two external testing datasets.

Supplementary Fig. 3. The receiver operating characteristic (ROC) curves of our UIOS+thresholding and other uncertainty-based methods+thresholding in internal and two external testing datasets.

3. The performance in terms of AUROC as shown in Fig. 2. In terms of the average AUROC across all classes, the performance difference between the standard AI model, UIOS and UIOS+threshold is very small for the internal test set (0.9968 vs 0.9979 vs 0.9989) and CJSIEC set (0.9767 vs 0.9907 vs 0.9977). A statistical significance test based on the DeLong's test needs to be performed to show that the difference in performance is statistically significant. Similar tests with p-values should also be reported for the benchmark methods (see point II above).

REPLY: Thanks for the helpful suggestions. In the revised manuscript, we further analyzed the statistical significance for the improvement of our proposed UIOS compared to the baseline model and other uncertainty-based approaches. The P-values of AUC for different datasets are shown in Supplementary Table 8, and the P-values of F1 scores on different datasets are given in Supplementary Table 7. The performance improvement regarding the AUC metric is significant on most datasets (Supplementary Table 8). In addition, UIOS achieved more evident improvement significance of F1 score compared to standard AI model and other uncertainty-based methods in most conditions.

Supplementary Table 7: P-values of F1 scores for UIOS model compared to other methods on different datasets

Methods	Internal testing set	TC-JSIEC	TC-unseen dataset	Internal testing set+thresholding	TC-JSIEC+thresholding	TC-unseen dataset+thresholding
UIOS->Baseline	0.029	0.006	0.008	/	/	/
UIOS->MC-Drop	0.012	0.049	0.001	0.005	0.091	0.004
UIOS->Ensemble	0.009	0.388	0.001	0.059	0.286	0.009
UIOS->CE	0.004	0.047	0.001	0.001	0.046	0.027
UIOS->TTA	0.003	0.030	0.001	0.006	0.048	<0.001

Supplementary Table 8: P-values of AUROC for UIOS model compared to other methods on different datasets

Methods	Internal testing set	TC-JSIEC	TC-unseen dataset	Internal testing set+thresholding	TC-JSIEC+thresholding	TC-unseen dataset+thresholding
UIOS->Baseline	0.371	0.002	0.185	/	/	/
UIOS->MC-Drop	0.040	0.049	0.003	0.066	0.132	0.008
UIOS->Ensemble	0.036	0.223	0.036	0.071	0.276	0.033
UIOS->CE	0.565	0.019	0.610	0.263	0.029	0.259
UIOS->TTA	0.020	0.003	0.007	0.048	0.042	0.004

Minor Comments:

In terms of implementation details in Sec 3.9

1. Why is only random flipping used for data augmentation? Typically, small rotation about the image center, color jitter, random resize-crop are also used while training deep learning models to improve generalization.

REPLY: Thanks for your question. Random flipping can augment the data without damaging to the pathological features of fundus diseases in the fundus images. Clinically, the diagnosis of fundus diseases is also based on morphology, color, and area size of the lesions. Other typical data augmentation methods such as small rotation about the image center, color jitter, and random resize-crop, may lead to variations in pathological features in some fundus diseases, resulting in data and label mismatches. Therefore, in this study, we only adopted random flipping used for data augmentation.

2. Is the GPU mentioned DGX 3090 correct or should it be Nvidia Geforce RTX 3090 ? I wanted to confirm as I have not heard of the DGX 3090 GPU before.

REPLY: Thanks for your kind reminder. It should be Nvidia Geforce RTX 3090, we have made careful corrections in the revised manuscript.

REVIEWERS' COMMENTS

Reviewer #1 (Remarks to the Author):

Thank you for the further clarifications and improvements to the manuscript. The revised manuscript is much improved in terms of clarity, precision, and the appropriate discussion of limitations. I think the authors have largely addressed my concerns.

I just have a few remaining minor concerns:

1. Regarding the point about why the UIOS model outperforms the standard AI model even on the "Target" diseases, and the fairness of the comparator arm – the authors have clarified that the UIOS model and the standard AI models were trained with different classifiers and different loss functions. I must admit that the technical details of this are outside of my area of expertise, but as long as the standard AI model was provided with a similar level of optimization, and is truly a fair comparator arm to the UIOS model, then the results here are valid.

2. Regarding the point about the UIOS model and the UIOS + thresholding model - I am glad that the authors have clarified in their rebuttal that all datasets and tasks used the exact same optimal threshold, which can be applied directly for clinical translation. This is ideal. I assume that this is the "UIOS + thresholding" model. However, it is still unclear to me what the separate "UIOS" model (without thresholding) indicates. I can only see how this model works in providing an uncertainty prediction when there is a threshold applied. Does that mean that the "UIOS" model alone only provides a diagnostic classification result, but does not provide any uncertainty prediction at all?

3. Regarding the point about how the diagnostic performance metrics in the manuscript are calculated when images are flagged up as uncertain by the UIOS model – the authors clarify in the rebuttal that when an image is flagged as "uncertain" beyond the threshold by the UIOS model, those images are removed when calculating the eventual diagnostic performance metrics. They mention that this has been clarified in the manuscript, but I am unable to find this section in the revised manuscript. Please double-check that this has been included and clarified, as it is an important point.

Reviewer #2 (Remarks to the Author):

Overall, the authors have answered all my comments in the previous round of the review process satisfactorily and have significantly improved the manuscript with the required additional experiments.

In my opinion the manuscript is suitable for acceptance.

Response to Reviewers' Comments

Paper ID: NCOMMS-23-15702A

We thank the associate editor and the reviewers for their excellent comments and suggestions which have helped us to greatly improve the quality of the paper. We have addressed all issues raised by the editor and reviewers and have revised the manuscript accordingly. The changes are marked in red font in the resubmitted version to facilitate the review process. In the response texts below, all references to figures, pages and citations correspond to the resubmitted version.

Reviewers #1 (Remarks to the Author):

Thank you for the further clarifications and improvements to the manuscript. The revised manuscript is much improved in terms of clarity, precision, and the appropriate discussion of limitations. I think the authors have largely addressed my concerns. I just have a few remaining minor concerns:

REPLY: We would like to thank you for your professional review work and valuable suggestions on our manuscript. As you are concerned, there are several minor issues need to be addressed, which are replied in detail as below. Thank you for your time and consideration.

1. Regarding the point about why the UIOS model outperforms the standard AI model even on the "Target" diseases, and the fairness of the comparator arm – the authors have clarified that the UIOS model and the standard AI models were trained with different classifiers and different loss functions. I must admit that the technical details of this are outside of my area of expertise, but as long as the standard AI model was provided with a similar level of optimization, and is truly a fair comparator arm to the UIOS model, then the results here are valid.

REPLY: Thank you for your comments and consideration. We appreciate your acknowledgment of the clarification we provided regarding the differences between the UIOS model and the standard AI model in terms of classifiers and loss functions. We assure you that the standard AI model was indeed optimized to a similar level to ensure a fair comparison with the UIOS model. The thoroughness of our experimental setup aimed to establish a valid and meaningful comparison between these models.

2. Regarding the point about the UIOS model and the UIOS +thresholding model - I am glad that the authors have clarified in their rebuttal that all datasets and tasks used the exact same optimal threshold, which can be applied directly for clinical translation. This is ideal. I assume that this is the "UIOS + thresholding" model. However, it is still unclear to me what the separate "UIOS" model (without thresholding) indicates. I can only see how this model works in providing an uncertainty prediction when there is a threshold applied. Does that mean that the "UIOS" model alone only provides a diagnostic classification result, but does not provide any uncertainty prediction at all?

REPLY: The "UIOS" model (without thresholding) also provides uncertainty scores, but not excludes images with uncertainty scores above the threshold for analysis. The UIOS model (without thresholding) were mainly used to indicate that UIOS can provide predictive results with uncertainty scores for reference and achieve comparable diagnostic performance to standard AI model.

3. Regarding the point about how the diagnostic performance metrics in the manuscript are calculated when images are flagged up as uncertain by the UIOS model – the authors clarify in the rebuttal that

when an image is flagged as “uncertain” beyond the threshold by the UIOS model, those images are removed when calculating the eventual diagnostic performance metrics. They mention that this has been clarified in the manuscript, but I am unable to find this section in the revised manuscript. Please double-check that this has been included and clarified, as it is an important point.

REPLY: Thanks for you kindly reminder. We have carefully proofread the manuscript and clarified this point in the revised submission, which has been highlighted in red. (Line 3-5, Page 4).

Reviewers #2 (Remarks to the Author):

Overall, the authors have answered all my comments in the previous round of the review process satisfactorily and have significantly improved the manuscript with the required additional experiments. In my opinion the manuscript is suitable for acceptance.

REPLY: Thanks for your time and encouragement.